# Extensive trimming of short single-stranded DNA oligonucleotides during replication-coupled gene editing in mammalian cells

**Thomas W. van Ravesteyn**[ID]**, Marcos Arranz Dols, Wietske Pieters, Marleen Dekker, Hein te Riele**[ID]*

Division of Tumor Biology and Immunology, The Netherlands Cancer Institute, 1066 CX Amsterdam, the Netherlands

* h.t.riele@nki.nl

**Data Availability Statement:** All relevant data are within the manuscript and its Supporting Information files.

## Abstract

Through transfection of short single-stranded oligodeoxyribonucleotides (ssODNs) small genomic alterations can be introduced into mammalian cells with high precision. ssODNs integrate into the genome during DNA replication, but the resulting heteroduplex is prone to detection by DNA mismatch repair (MMR), which prevents effective gene modification. We have previously demonstrated that the suppressive action of MMR can be avoided when the mismatching nucleotide in the ssODN is a locked nucleic acid (LNA). Here, we reveal that LNA-modified ssODNs (LMOs) are not integrated as intact entities in mammalian cells, but are severely truncated before and after target hybridization. We found that single additional (non-LNA-modified) mutations in the 5'-arm of LMOs influenced targeting efficiencies negatively and activated the MMR pathway. In contrast, additional mutations in the 3'-arm did not affect targeting efficiencies and were not subject to MMR. Even more strikingly, homology in the 3'-arm was largely dispensable for effective targeting, suggestive for extensive 3'-end trimming. We propose a refined model for LMO-directed gene modification in mammalian cells that includes LMO degradation.

## Author summary

The first step of many gene editing approaches in mammalian cells is to generate a targeted DNA lesion. By administering a repair template as second step, endogenous DNA repair mechanisms can be misled to introduce specific gene variants. However, subtle gene modification can also be achieved with high precision through a one-action protocol in the absence of DNA breaks. We have shown before that short single-stranded DNA molecules (LMOs) are very useful to introduce and study genetic variants that may predispose patients to cancer. While LMOs are known to integrate into the genome during DNA replication, the precise mechanism is poorly understood. We targeted mouse embryonic stem cells with differently designed LMOs to examine their effectiveness and editing outcomes. Based on these results we conclude that the two LMO termini are processed at different moments during the gene editing process. While the 3'-arm is degraded

**Funding:** This work was financially supported by the Dutch Organization for Scientific Research, URL https://www.nwo.nl/ [grant numbers ALW 822.02.01, TTW 14888 to HTR]. The funders had no role in study design, data collection and analysis, decision to publish, or preparation of the manuscript.

**Competing interests:** The authors have declared that no competing interests exist.

prior to LMO binding to the target site, the 5'-arm is degraded afterwards. Counterintuitively we also observe that partial degradation of the 3'-arm increases targeting efficiencies. Taken together our data provides novel mechanistic insight into our understanding of replication-coupled gene editing and may guide future LMO design strategies.

## Introduction

The ability to generate gene modifications is of great importance to a wide variety of research fields in molecular biology. Especially the ability to generate precise gene modifications at endogenous loci with the resolution of single nucleotides enables the study of specific protein residues. Various strategies have been developed to edit the genome with single-stranded repair templates in combination with site-specific nucleases such as Zinc-finger nucleases [1], TAL-effector nucleases (TALENs) [2] or CRISPR/Cas9 [3]. Besides use as repair-template in combination with a site-specific DNA double stranded break (DSB), ssODNs with a centrally positioned mutation are also used to generate subtle gene modifications in the absence of DSBs. Targeting chromosomal DNA during replication has proven to be highly effective for multiplex genome engineering in simple prokaryotic and eukaryotic model organisms like *Escherichia coli* [4–6] and *Saccharomyces cerevisiae* [7,8]. In addition, we have demonstrated the applicability of this technology in mammalian cells by setting up screens that enable the classification of Lynch syndrome-associated variants of uncertain clinical significance in MSH2, MSH6 and MLH1 [9–11].

Over the years different mechanistic models have been proposed for the process by which ssODNs integrate into the genome of mammalian cells (reviewed by Aarts and te Riele: [12]), but most evidence suggests that the process takes place during DNA replication [13–16]. According to this model ssODNs hybridize to their target site when single-stranded DNA is exposed at the replication fork due to the unwinding of the DNA double helix by replicative helicases. Thereafter ssODNs may prime DNA synthesis by replicative polymerases. Finally, ssODNs become physically integrated into the genome and thereby introduce mutations to the nascent DNA [17,18].

It has become evident that MMR greatly suppresses targeting efficiencies in both eukaryote and prokaryote organisms by 2–3 orders of magnitude [7,19,20]. In eukaryotic cells heterodimeric protein MutSα is involved in the detection of base-base mismatches that are the result of replication errors [21]. Together with MutLα it initiates a repair process that leads to excision of the nascent strand containing the falsely incorporated nucleotide. Thereafter replicative polymerases get a second opportunity to resynthesize the DNA. Similarly, also mismatches arising from annealing of the ssODN to its chromosomal complement elicit a MMR reaction, which in this case leads to abortion of the gene modification reaction. Recently, we have found that MMR can be evaded specifically at the site of modification by the inclusion of a single locked nucleic acid (LNA) at the central mismatching nucleotide of ssODNs in mouse embryonic stem cells (mESCs) and *E. coli* [22]. LNA modification of ssODNs prevented MMR activation and led to equal targeting efficiencies in MMR-deficient (MMR⁻) and -proficient (MMR⁺) cells. Subsequent optimization of the protocol resulted in targeting efficiencies in the order of $10^{-3}$ in MMR⁺ mESCs. The possibility to evade MMR, the simplicity of the protocol (no additional components are required) and its high precision make it an attractive alternative for the generation of a large variety of subtle gene modifications, especially if they result in a selectable phenotype [9,22].

To study the processes that affect the integration of LMOs into the genome in more detail, we made use of LMOs designed to correct the defective AAG start codon of a neomycin (*neo*) reporter in mESCs but containing one or more additional mismatch (AMM)-creating nucleotides. We found that the presence of single AMMs in the 5'-arm or 3'-arm of LMOs differentially influenced the outcome of gene modification in terms of targeting efficiencies, activation of the MMR pathway and introduction of the AMM into the genome. Based on these data we propose a refinement of the current replication-dependent model for ssODN-directed gene modification in mammalian cells: LMOs are not incorporated as intact entities but undergo several trimming events before they become incorporated into the genome of mESCs.

## Results

### LMOs with single additional mismatches reveal incomplete integration of LMO sequences

To study and optimize the LMO-directed gene modification procedure in mammalian cells we made use of a single copy *neo*-reporter, stably integrated at the *Rosa26* locus in MMR$^+$ and MMR$^-$ mESCs [19]. Successful restoration of the defective start codon (AAG) into a functional start codon (ATG) by LMOs with a length of 25 nucleotides (nt), results in G418-surviving colonies of which the number reflects the targeting efficiency [22]. Correction of this *neo* reporter is achieved with 25 nt LMOs at an efficiency of 1 x 10$^{-3}$ in MMR$^+$ cells [22]. To study LMO integration in the absence of MMR we used *Msh2*$^{-/-}$ mESCs [19] in which MMR is fully abrogated (which is not the case in *e.g. Msh6*$^{-/-}$ cells [23]). In some experiments, we used a reporter in which *neo* was replaced for *Gfp*, allowing the targeting efficiency to be monitored by flow cytometry.

We individually transfected six LMOs containing a single additional mismatch (AMM) at nucleotide position (p)1, p5, p9, p17, p21 or p25 into MMR$^+$ and MMR$^-$ mESCs and determined their targeting efficiencies (Fig 1A and 1B). In the MMR$^-$ cell line, we observed that the efficiency was moderately reduced by an AMM in the 5'-arm at p1, p5 or p9, whereas an AMM in the 3'-arm at p17, p21 or p25 yielded efficiencies equal to the control LMO (ctrl.). By comparing the normalized efficiencies in MMR-deficient and -proficient cells, we found that the presence of MMR exacerbated the suppressive effect of 5'-arm AMMs at p5 (2.7 fold) and p9 (>80-fold) (Fig 1B). Consistently, the extra suppression by MMR was prevented through LNA modification of these AMMs (S1 Fig). Strikingly, MMR did not affect targeting efficiencies when AMMs were present in the 3'-arm (Fig 1B).

To determine whether AMMs were co-introduced with the ATG-restoring mutation during LMO-directed gene modification we analyzed the *neo* locus of individually picked G418$^R$ colonies by Sanger sequencing (Fig 1C and 1D). In MMR$^-$ cells we found co-introduction of AMMs at positions 5, 9 and 17 in 11%, 100% and 21% of analyzed colonies, respectively. By contrast, no integration of AMM at p1, p21 and p25 was detected. Thus, even in the absence of MMR a considerable amount of LMO-encoded sequence information was lost during the targeting process. In MMR$^+$ cells none of the G418$^R$ colonies demonstrated integration of AMM at p1, p5, p17, p21 or p25 (Fig 1D). Only AMMp9, which greatly suppressed the targeting efficiency, was introduced in all colonies. These very few colonies probably arose in the rare event of a failed MMR response to AMMp9, resulting in incorporation of both AMMp9 and the central *neo* correcting mutation. In summary, the efficiency and sequencing data suggests that AMMp5 and -p9 were initially present in the annealed oligonucleotide, but then were recognized by MMR and subsequently excised, in most cases together with the LNA-protected central mismatch. In contrast, 3' AMMs resist MMR and do not affect the targeting efficiency, which may be indicative for 3'-end degradation prior to integration.

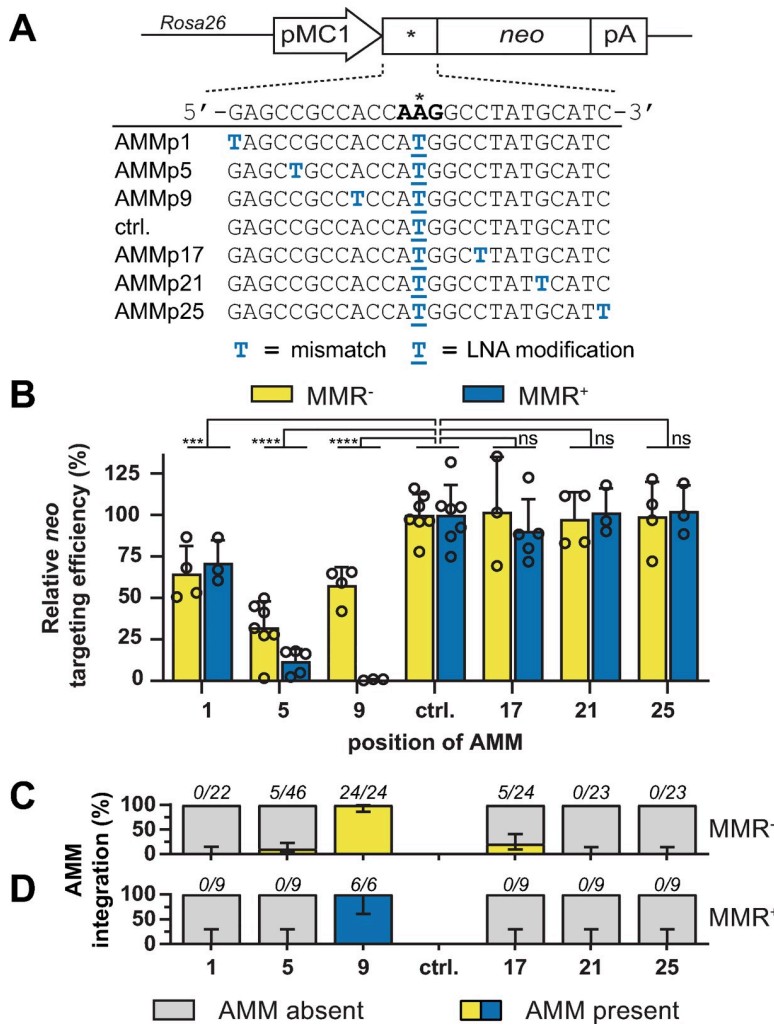

**Fig 1. Differential effects of single AMMs in the 5'- and 3'-arm of LMOs.** (**A**) Schematic representation of the stably integrated *neo* reporter in mESCs and sequences of LMOs that generate a functional *neo* start codon and contain single AMMs. Blue capital characters indicate mismatches with respect to the reporter, underlined characters indicate LNA modifications. (**B**) Relative *neo* targeting efficiency of LMOs with single AMMs at the indicated positions in MMR+ and MMR- cells. Bars indicate the mean with SD of at least three experiments. Significance was determined using a corrected two-way ANOVA. (**C, D**) Proportion of G418R colonies from MMR- (C) and MMR+ (D) cells in which the indicated AMM was integrated as determined by Sanger sequencing. Error bars represent 95% confidence interval.

To corroborate these conclusions for a different endogenous locus, we targeted MMR+ *Msh2*+pur/Δ mESCs in which one *Msh2* allele (Δ) is fully deleted [9]. LMOs were designed to generate MSH2 loss-of-function variant P622L and contained an AMM at p5, 9, or 17 (S2A Fig). To mimic MMR-deficiency and proficiency, we used LNA-modified or non-modified AMMs, respectively. These LMOs also carried a 5'-terminus 6-chloro-2-methoxyacridine (Acr) modification that enhances targeting efficiencies [22,24]. Successful introduction of P622L results in resistance to the methylating agent 6-thioguanine (6TG) [9] enabling quantification of the targeting efficiency by counting the number of 6TGR colonies. Consistent with our conclusion that the 5'-arm is present during target hybridization, AMMp5 and -p9 triggered a MMR response which resulted in a severely reduced targeting efficiency (S2B Fig). Inclusion of a MMR-evading LNA at p5 or p9 partially negated the strong efficiency reduction. In contrast to LNA-modified AMMp9, LNA-modified AMMp5 was introduced only in a

minority of successful targeting events (S2C Fig), suggestive for a MMR-independent mechanism of 5'-arm degradation after annealing. Consistent with 3'-arm degradation before target hybridization, AMMp17 did not trigger MMR and did not influence the targeting efficiency. LNA-modified AMMp17 incorporated at a rate of 75%, albeit with a mild efficiency reduction. Again these results indicate that only a small part of the LMO becomes stably integrated into the genome.

## Sequence composition of the 3'-arm is largely irrelevant for LMO-mediated gene modification

In addition to LMOs with a single AMM, we also targeted MMR⁺ and MMR⁻ cells with LMOs containing multiple AMMs. Whereas three 5'-arm mismatches synergistically suppressed the targeting efficiency, three AMMs in the 3'-arm only moderately affected the targeting efficiency (Fig 2A and 2B). Consistently, in case of concomitant 5'- and 3'-arm AMMs, targeting efficiencies were largely determined by the 5'-arm AMMs and decreased with increasing number of AMMs (S3A and S3B Fig). Moreover, sequencing analysis revealed that AMM incorporation follows the same trend as for LMOs with a single AMM (S3C, S3D, S4A and S4B Figs). Similar results were obtained with LMOs carrying 5'-Acr modification (S3E–S3H, S4C and S4D Figs).

Taken together, 5'-arm located AMMs reduced targeting efficiencies and were prone to MMR response. This indicates that the 5'-arm remained part of the LMO during targeting and was likely subject to degradation after target hybridization. By contrast, LMOs with AMMs residing in the 3'-arm demonstrated equal-to-control efficiencies and did not evoke a MMR response that suppressed the efficiency, suggesting that the 3'-arm was partially degraded before annealing with its target. Strongly supportive for 3'-end degradation prior to annealing, we found the nucleotide composition of the 3'-arm to be largely irrelevant. Replacing the nine 3'-terminal nucleotides for a stretch of nine C, A or T nucleotides or for arbitrary sequences or random nucleotides had minimal effect on *Gfp* targeting efficiencies (Figs 2C, 2D, S3I and S3J). Also at a different locus, *Msh2*, 3'-terminal C, A and T stretches did not affect the targeting efficiency (S3K and S3L Fig). These results strongly indicate these nucleotides were removed prior to annealing. A stretch of G nucleotides, however, strongly suppressed efficiencies, which may be due to their tendency to form nuclease-resistant secondary structures.

## 5'-arm of LMOs is degraded after target hybridization by an endonuclease

As we found that the sequence dependence and degradation of LMOs is different for the 5'- and 3'-arm, we investigated these processes separately. From previous work it has become evident that targeting efficiencies can be increased by protecting ssODNs from nucleolytic degradation by modifying the 5'-terminus of a ssODN with LNA or Acr [22,24–26]. To investigate the influence of these modifications on AMMp5 integration we transfected LMOs with AMMp5 and additional LNA or 5'-Acr modification to MMR⁺ and MMR⁻ cells and determined the efficiency of targeting (Fig 3A–3C). As expected, 5'-end modification increased the targeting efficiency of control and AMMp5 LMOs in both MMR⁻ and MMR⁺ cells (Fig 3B and 3C; please note the y-axis dimensions are different as AMMp5 reduces targeting efficiency). However, the targeting-suppressive effect of AMMp5 in MMR⁻ and even more in MMR⁺ cells was not affected by LNA at p1 or 5'-Acr modification of the LMO (Fig 3C). Moreover, sequencing revealed that LNA at p1 or 5'-Acr protection hardly affected the incorporation of AMMp5 (Fig 3D and 3E). Since 5'-terminus modification by LNA or Acr increased targeting efficiencies, possibly by providing protection from exonucleolytic degradation or through increased stability of the LMO-target heteroduplex, the mostly unaffected incorporation rate

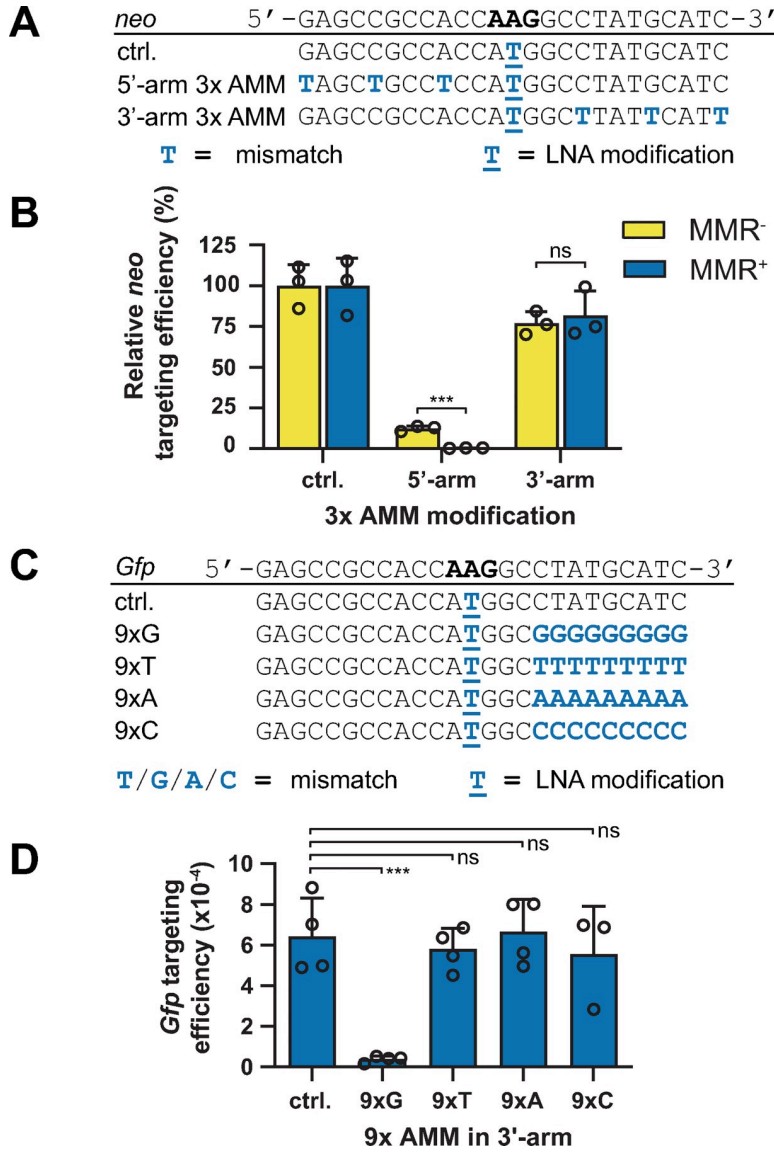

**Fig 2. Differential effects of multiple AMMs in the 5'- and 3'-arm of LMOs.** (**A**) Sequence of LMOs with three AMMs in the 5'- or 3'-arm. Blue capital characters indicate mismatches with respect to the reporter, underlined characters indicate LNA modifications. (**B**) Relative *neo* targeting efficiency of LMOs with three AMMs in MMR⁻ and MMR⁺ cells. Bars indicate the mean with SD of three experiments. Significance was determined using a corrected multiple t-test. (**C**) Sequence of LMOs with consecutive tracts of nine identical bases in the 3'-arm. (**D**) Targeting efficiency at the *Gfp* reporter of LMOs containing mononucleotide tracts in the 3'-arm.

of AMMp5 suggests that LMO annealing is eventually followed by removal of 5'-arm nucleotides through endonucleolytic activity.

During regular DNA replication, Flap endonuclease 1 (FEN1) is recruited to remove the flap structure from the 5'-end of a downstream Okazaki-fragment [27]. As LMOs are thought to hybridize to the target site during DNA replication at the lagging strand template, we hypothesized that FEN1 could be involved in 5'-arm LMO degradation. By lentiviral transduction of two different *Fen1*-shRNAs we generated stable FEN1 knockdown (KD) clones in MMR⁻ mESCs that showed strong reduction of FEN1 protein levels (S5A and S5B Fig). However, the targeting efficiencies of both the control and AMMp5 LMO in two FEN1 KD clones

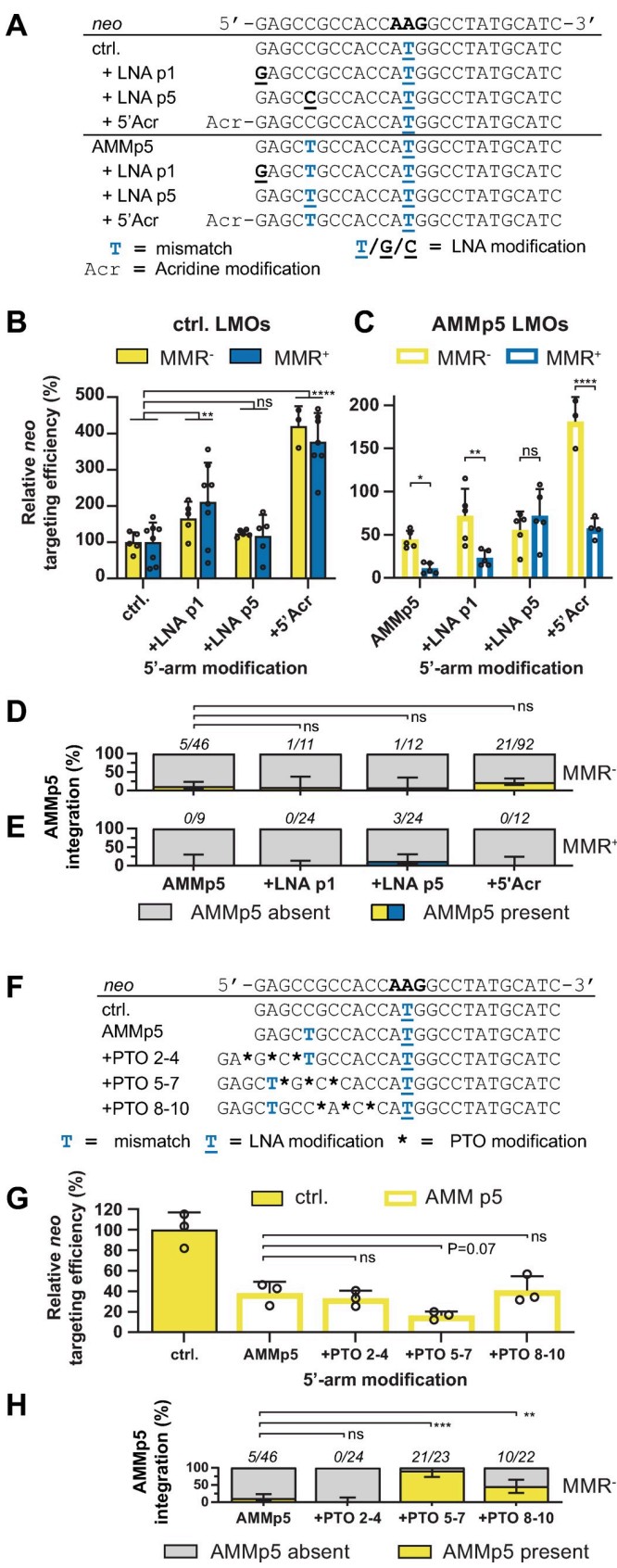

**Fig 3. The 5'-arm of LMOs is degraded during the process of targeting through endonuclease activity.** (**A**) Sequence of 5'-arm modified LMOs with and without AMMp5. Blue capital characters indicate mismatches with respect to the reporter, underlined characters indicate LNA modifications, Acr indicates modification at 5'-terminus with 6-chloro-2-methoxyacridine. (**B, C**) Relative *neo* targeting efficiency of 5'-arm modified control LMOs (B) and AMMp5 LMOs (C) in MMR⁻ and MMR⁺ cells normalized to control LMOs. Bars indicate the mean with SD of at least three experiments. Significance was determined using a corrected two-way ANOVA. (**D, E**) Proportion of G418ᴿ colonies from MMR⁻ (D) and MMR⁺ (E) cells in which AMMp5 was integrated as determined by Sanger sequencing. (**F**) Sequence of AMMp5 LMOs with PTO modifications in the 5'-arm. Asterisks indicate PTO bonds. (**G**) Relative *neo* targeting efficiency of PTO-modified LMOs with AMMp5 in MMR⁻ cells. Bars indicate the mean with SD of three experiments. Significance was determined using a corrected one-way ANOVA. (**H**) Proportion of G418ᴿ colonies from MMR⁻ cells in which AMMp5 was integrated as determined by Sanger sequencing. Error bars for AMMp5 integration rate (C, D, G) represent 95% confidence interval.

were similar as in the parental cell line (S5C Fig). Also the integration rate of AMMp5 was highly similar in the FEN1-KD clones and the parental cell line (S5D Fig). Possibly, endonucleolytic processing by FEN1 or another endonuclease is a necessity for every LMO integration. Consistently, providing a LMO with a 5'-phosphate that could be ligated directly to a free 3'-end did not augment gene modification efficiency (S5E Fig) and did not affect incorporation of AMMp5 (S5F Fig), indicative for endonucleolytic activity.

To further examine 5'-arm degradation, we transfected MMR⁻ cells with AMMp5 LMOs that were modified with three consecutive nuclease-resistant phosphothioate (PTO) bonds at different positions (Fig 3F and 3G). We observed that PTO bonds at the 5'-terminus had no effect on the incorporation of AMMp5, suggestive for endonucleolytic cleavage downstream of AMMp5. By contrast, internal PTO modification between p5 and p8 increased the AMMp5 incorporation rate from 11% up to 91% (Fig 3H). Internal PTO bonds between p8 and p11 also increased the incorporation rate but more modestly to 45%, consistent with available endonucleolytic cleavage sites on both sides of AMMp5. These data demonstrate that integration of 5'-arm mutations can be stimulated by internal, but not by terminal PTO modification, thus indicating that 5'-arms of LMOs are indeed processed by endonucleolytic activity.

## Effective protection from 3'-arm degradation reduces LMO targeting efficiencies

Next, we investigated the effect of preventing degradation that takes place at the 3'-arm. We assessed the effects of Acr, LNA, 2'O-Me and PTO modifications in the 3'-arm of AMMp17 ssODNs in MMR⁻ cells (S6 Fig). We observed that LNA, 2'O-Me and three PTO modifications did not significantly affect targeting efficiencies, while internal Acr reduced the targeting efficiency (S6B Fig). Whereas internal Acr did not affect the incorporation rate of AMMp17, sequencing revealed that LNA, 2'O-Me and PTO modifications increased the incorporation rate for AMMp17 from 42% to 100% (S6C Fig). Thus, degradation of the 3'-arm was effectively prevented by these three modifications, without affecting the targeting efficiency.

We next combined LNA modification in the 3'-arm with the central mismatching LNA in control and AMMp17 LMOs (Fig 4A–4C). Again we found that a LNA located in the 3'-arm at p17 or p21 increased the integration rate of AMMp17 from 21% to ~70% in MMR⁻ cells (Fig 4D), indicative for reduced 3'-arm degradation. Notably, LNA modification of the 3'-terminal position (p25) did not affect AMMp17 incorporation rate. In MMR⁺ cells AMMp17 could only be integrated when an LNA was placed at the same position as the extra mismatch (Fig 4E). While the increased incorporation rate of AMMp17 in MMR⁻ cells indicates that 3'-arm degradation can be effectively prevented, the positioning of a second LNA on p21 adversely affected the efficiency by 75%. This suggests that gene modification is promoted by 3'-arm degradation.

In an alternative attempt to reduce the level of 3'-arm degradation we generated knockout clones for TREX1 (S7A Fig), one of the most abundant 3'-5' exonucleases in the cell that acts

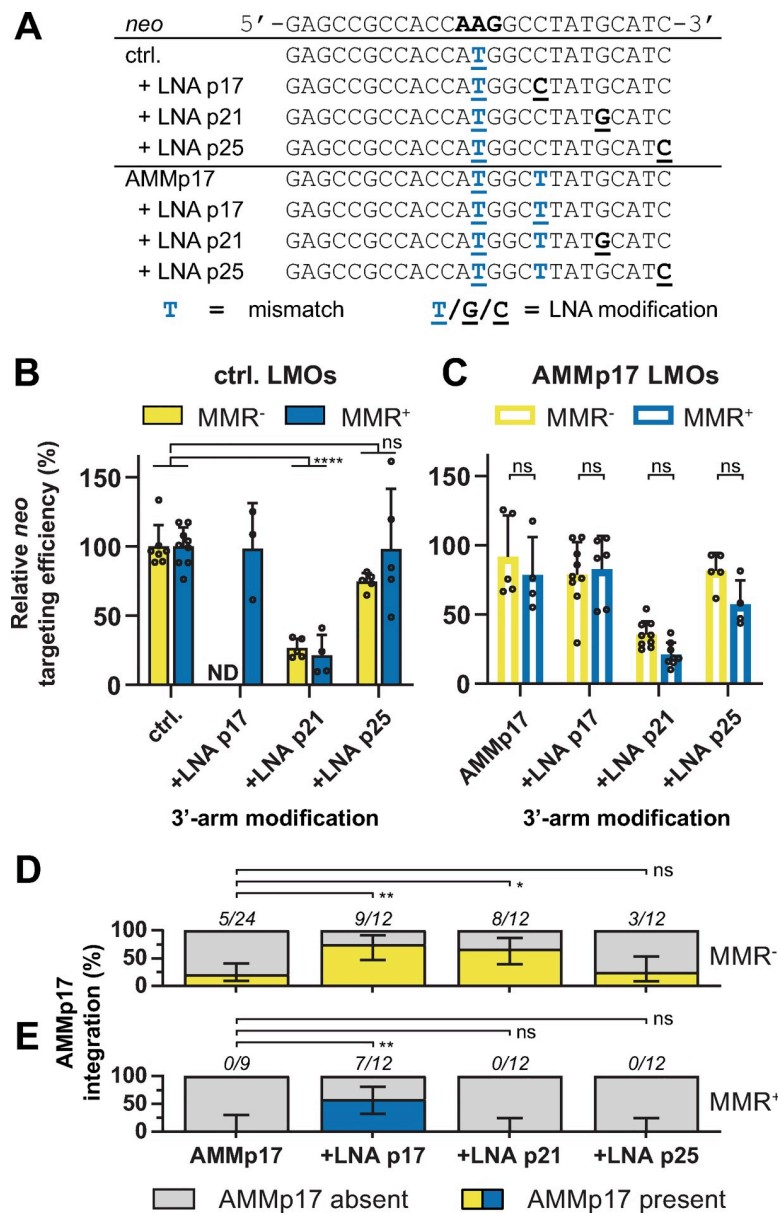

**Fig 4. Suppression of 3'-5' LMO degradation increases 3'-arm integration but does not increase targeting efficiencies.** (**A**) Sequences of control and AMMp17 LMOs with a second LNA modification in the 3'-arm. Blue capital characters indicate mismatches with respect to the reporter, underlined characters indicate LNA modifications. (**B, C**) Relative *neo* targeting efficiency of 3'-arm modified control LMOs (B) and AMMp17 LMOs (C) in MMR⁻ and MMR⁺ cells normalized to control LMO. Bars indicate the mean with SD of at least four experiments. ND indicates not determined. Significance was determined using a corrected two-way ANOVA. (**D, E**) Proportion of G418ᴿ colonies from MMR⁻ (D) and MMR⁺ (E) cells in which AMMp17 was integrated as determined by Sanger sequencing. Error bars represent 95% confidence interval.

on single-stranded DNA [28–30]. Surprisingly, in comparison to the parental cell line we obtained lowered efficiencies in all of four analyzed TREX1 KO clones (S7B Fig). In line with results observed for LMOs with a second LNA on p21, this suggests that 3'-5' degradation is beneficial for effective targeting with LMOs. Endogenous 3'-5' exonuclease activity is probably sufficient in mESCs as overexpression of hTREX1 did not influence targeting efficiencies (S7C–S7E Fig).

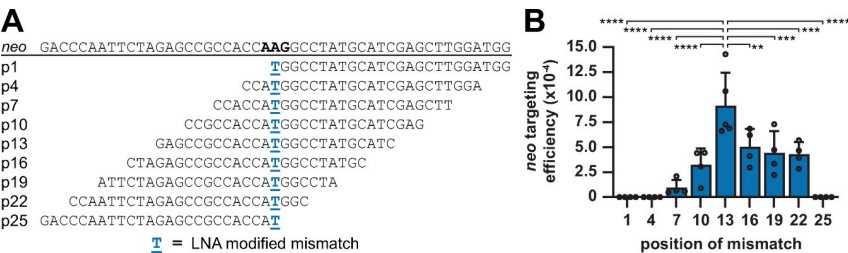

**Fig 5. LMOs with a centrally placed LNA-protected mismatch provide the optimal targeting efficiency. (A, B)** Sequence (A) and *neo* targeting efficiency in MMR[+] cells (B) of LMOs with repositioned LNA-protected mismatch. Blue underlined capital characters indicate LNA-protected mismatch. Bars indicate the mean with SD of four experiments. Significance was determined using a corrected one-way ANOVA.

## LMOs with a centrally positioned LNA-protected mismatch yield the highest targeting efficiency

The finding that LMOs are subject to extensive nucleolytic degradation prompted us to determine the optimal position of the LNA-protected mutation in a 25 nt LMO (Fig 5). We confirmed that placement of the mutation in the central position yielded the highest efficiency. Consistent with 5'-end degradation through endonucleolytic activity, we observed that positioning the mutation towards the 5'-end decreased the efficiency profoundly. A more subtle efficiency decrease was detected for LMOs with the mutation placed towards the 3'-end. Since this subtle efficiency decrease seems to contradict the extensive degradation of the 3'-arm as observed above, we analyzed the protective effect of LNA modification in this repositioning experiment. By comparing the targeting efficiency of ssODNs with and without LNA protection at the mutating position in MMR[-] cells we found that LNA modification increased the targeting efficiency if the mutation was central or repositioned towards the 3'-arm at p19 (S8 Fig). No benefit from LNA was observed when the mutation was positioned in the 5'-arm. Taken together, we envisage that the complex interplay between 5'- and 3'-arm degradation leads to an outcome in which a centrally placed mutation has the highest likelihood to remain unaffected by both degradation processes. In addition to MMR evasion, it appears that LNA modification of this central mutation enhances the targeting efficiency by providing protection from degradation.

## Discussion

Our work provides new mechanistic insight into LMO-directed gene modification at the replication fork in mammalian cells. In summary, by incorporation of additional mutations into short 25 nt LMOs we found evidence for extensive degradation of the 5'- and 3'-arm of LMOs during targeting (Fig 6). We show that mutations in the 5'-arm strongly reduce the targeting efficiency and demonstrate that their presence can trigger a MMR response. Apparently, these mutations remain part of the LMO during target hybridization and are subject to MMR surveillance. Inclusion of 5'-end protection by LNA or Acr increases targeting efficiencies, but in the majority of integration events the first 6 to 8 nt of the 5'-arm were removed by endonucleolytic activity. As MMR exacerbated the suppressive effect of 5'-arm AMMs at p5 and p9, the single LNA modification at the central mismatch was apparently insufficient to block nucleolytic degradation upon activation of MMR by adjacent mismatches. Alternatively, MMR-directed 5'-arm loss may destabilize annealed LMOs and hence prohibit gene correction.

In contrast to the 5'-arm, we demonstrate that the 3'-arm is likely to be degraded before target hybridization. Remarkably, inclusion of one or multiple mutations in the 3'-arm of LMOs barely influenced targeting efficiencies in MMR[+] cells. Consistently, it has been shown that

**Fig 6. Degradation steps during LMO-directed gene modification.** (**A**) After transfection to mammalian cells, LMOs are partially degraded by 3'-5' exonucleases; the centrally positioned LNA provides protection from degradation. (**B**) The 3'-arm truncated LMOs anneal to their ssDNA target site during DNA replication. (**C**) DNA MMR scans DNA for mismatches and removes the nascent strand carrying a non-matching base. LNA-protected mismatches evade MMR. (**D, E**) After annealing, the 5'-arm of the LMO is removed through endonucleolytic activity before it becomes fully integrated (E) into the newly synthesized DNA strand.

ssODNs with a 3'-H group were equally active as ssODNs with a 3'-OH terminus [17]. Based on these findings we have extensively tested various LMO designs of which some were able to block 3'-arm degradation. For example, a second mutation was efficiently introduced in MMR$^+$ cells through the incorporation of an additional LNA-protected mismatch at p17. Interestingly, through inclusion of an additional LNA at p21 or by removal of 3'-5' exonuclease TREX1, we unexpectedly found that degradation of the 3'-arm might be beneficial for optimal targeting efficiencies if a central LNA modification is present. Intriguingly, in the absence of a central LNA, the p21 LNA did not affect targeting efficiency (compare Figs 4C and S6B). We therefore propose that 3'-arm degradation can be beneficial, but becomes deleterious when degradation extends beyond the central mismatch. Thus, protection of a non-modified central mismatch by the p21 LNA compensates for the loss of otherwise beneficial 3'-arm degradation. When the central mismatch is already protected by an LNA, the negative effect of 3'-arm retention by p21 LNA dominates. Taken together, our data indicate that nucleases process LMOs and thereby provide a biologically active molecule that can integrate into the genome of proliferating cells [17,31].

We realize that our model for 5' and 3' processing of ssODNs was deduced from colony counts and sequencing data rather than based on direct experimental evidence. However, given the modest frequency of successful targeting events, it is uncertain whether the fate of the total ssODN population is representative for the few ssODNs that ultimately effectuated the desired gene modification. We therefore restricted our analyses to successful targeting events in order to obtain information on the fate of the "winning ssODN". Nevertheless, ssODN attributes such as stability, target site affinity, promiscuity and type of mismatches could affect intracellular processing and hence targeting efficacy. Therefore, we cannot exclude that ssODNs designed for other loci might be processed differently yielding unanticipated results. Nevertheless, most data were obtained by comparing highly similar ssODNs targeting the defective start codon of an integrated *neo/Gfp* reporter and the relaxed requirement for 3'-arm homology was validated at the *Msh2* locus.

Although replication-coupled gene editing using LMOs is highly precise at the target locus, we realize that the use of short LMOs in combination with intracellular degradation processes may increase off-target gene editing rates. While the sequence of the 3'-arm was found to be largely irrelevant for targeting, we observed that mutations within the 5'-arm activate MMR (Figs 1, S1 and S2). Hence, if a LMO anneals to an off-target site and thereby generates one or more mismatches at the 5'-arm, MMR would strongly suppress off-target LMO integration. To minimize potential off-target LMO integrations we recommend to apply replication-coupled gene editing solely in MMR$^+$ cells.

Direct modification of genomic DNA at the replication fork has proven to be highly effective in non-mammalian cells like *E. coli* and *S. cerevisiae*. Efficient targeting in *E. coli* requires

ssODNs of at least 35–60 nt in length to allow for binding by phage protein Beta which provides additional protection from degradation and promotes hybridization to transiently single-stranded target regions [32–35]. Nevertheless, *E. coli* also showed a pattern of incomplete ssODN incorporation with decreasing incorporation rates towards the termini [36]. In addition, homology at the 5'-arm was found to be of much more importance than at the 3'-arm [37]. Similar to our 5'-Acr modified LMOs, 5'-PTO modification enhanced targeting efficiencies in *E. coli* while 5'-end degradation by DNA Pol I was not affected [4,36,37].

Previous observations in *S. cerevisiae* also indicated loss of ssODN sequences during the process of targeting [8,38]. Only bases within a central core of about 15 out of 40 nt ssODNs were incorporated regularly [38]. While the presence of MMR reduced targeting efficiencies, no effect of MMR was detected on integration patterns in *S. cerevisiae*. Furthermore, it was found that elimination of FEN1 in *S. cerevisiae* led to an incorporation rate increase specific for mutations which were included in the 5'-arm at both leading and lagging strand [38]. In combination with our observations in mammalian cells, this suggests a highly conserved mode of 5'-end processing of ssODNs at the replication fork. Given the pivotal role of FEN1 in Okazaki fragment maturation during DNA replication in mammalian cells [27], FEN1 is a likely candidate for 5'-arm excision. However due to its essential function and intricate regulation we haven't been able to conclusively determine its role in LMO degradation. Besides or instead of FEN1, other structure-specific endonucleases that are active during DNA replication may be involved in LMO degradation [39].

In *E. coli*, removal of a set of five exonucleases resulted in improved performance of Lambda Red-mediated multiplex genome engineering by ssODNs [40]. However, singleplex engineering benefitted only from removal of four exonucleases at 100-fold reduced ssODN concentrations [41], suggesting that only under conditions with limited ssODN concentrations, protection from degradation by exonuclease removal results in elevated efficiencies. At variance, we propose that 3'-arm degradation might be beneficial for targeting with LMOs as an additional LNA at position 21 in the 3'-arm, or knockout of exonuclease TREX1, reduced targeting efficiency in mESCs. Possibly, reduced efficiency of a LMO may occur when the 3'-arm can form a nuclease-resistant secondary structure. Poor performance of certain LMOs may be relieved by replacing the 3'-arm for a degradation-prone sequence like nine C, T or A nucleotides.

LMO-based targeting efficiencies of up to $3 \times 10^{-3}$ have been achieved for correction of the defective *neo* start codon using 5'-Acr modified LMOs [22]. This technology is currently less efficient than the ±40% gene editing frequencies that we have obtained for the same target site through templated-repair of a Cas9-induced DSB using 120 nt ssODNs [42]. However, the lack of additional planned and unforeseen mutations at the target site (which is adherent to CRISPR/Cas) makes LMO technology especially useful for direct phenotypic assessment of variants identified in disease-related genes. As a proof-of-concept we have demonstrated this approach for the classification of variants of uncertain clinical significance (VUS) identified in the MMR genes *MSH2* [9], *MSH6* [10] and *MLH1* [11]. While identification of pathogenic MMR gene variants was based on their acquired resistance to a methylating compound, the readout in other VUS screens could be based on more subtle and less binary phenotypic changes that can be assessed by flow cytometry, high-throughput microscopy or integrative single cell analyses [43,44]. Thus, while templated-repair of induced DSBs is an efficient approach to generate, and subsequently characterize gene variants with non-selectable phenotypes, LMO-based replication-coupled gene editing is useful to characterize gene variants that result in easily-detectable phenotypic changes.

In conclusion, we find that 25 nt LMOs are processed in mammalian cells before and after annealing to their chromosomal target sequence during replication-coupled gene modification. Upon successful targeting in mESCs we estimate that only ~8 out of the 25 nt of a LMO

become physically incorporated into the genome. This remarkable finding provides new mechanistic insights into a previously over-simplified model and may help to rationally design LMOs that incorporate multiple mutations simultaneously. We find that LNA protection of the centrally positioned mismatch not only prevents MMR activation in mESCs, but also 3'-5' degradation. Also in CHO and HeLa cells optimal efficiencies were achieved by protecting the central mismatch from degradation with nearby PTO modifications [45,46]. Based on our findings, better LMO design and the use of genetically adapted recipient cell lines may enhance targeting efficiencies which could result in broader application of this gene editing technique in mammalian cells.

## Materials and methods

### Cell culture and ssODN-directed gene modification of mESCs

To determine ssODN-directed gene modification efficiencies in MMR$^+$ and MMR$^-$ mammalian cells we made use of wild-type (MMR$^+$) and *Msh2$^{-/-}$* (MMR$^-$) mESCs with a single stably integrated *neo* or *Gfp* reporter [18,19]. mESCs were routinely cultured on top of a feeder layer of irradiated mouse embryonic fibroblasts (MEFs) in complete medium (CM) [47]. For experiments mESCs were cultured feeder-free in 60% Buffalo Rat Liver (BRL)-conditioned CM on gelatin-coated cell culture plastics. One day prior to targeting the *neo* reporter, 7 x 10$^5$ cells were seeded to 6-wells. Cells were transfected with 3 μg ssODN (unless otherwise indicated) in complex with 7.5 μL TransIT-siQUEST (Mirus) in 250 uL serum-free medium [22]. 24 h after transfection cells were counted on a CASY 1 cell counter (Schärfe System) and seeded to 8.5 cm plates in 30% BRL-conditioned CM. Next day cells were exposed to 750 μg/mL G418 (Geneticin; Life Technologies) to select for successfully targeted cells. The number of surviving colonies was determined 10–12 days after seeding to 8.5 cm plates. For targeting P622L in exon 12 of *Msh2* we made use of a previously published hemizygous mESC cell line (*Msh2$^{+pur/Δ}$*) [9]. Cells were seeded and targeted as described above and were seeded to 8.5 cm plates 2 days after targeting. One day thereafter we started selection for targeted cells using 300 μM 6-thioguanine (6TG) (Sigma-Aldrich). Number of resistant colonies was determined after 12 days. Efficiency represents the number of colonies over the total number of seeded cells. To simplify the comparison of targeting outcomes for different ssODNs across different cell lines, targeting efficiencies were normalized to control ssODN where indicated using the following method: for every single replicate efficiencies were divided by the mean targeting efficiency of the control ssODN per experiment and per cell line (Efficiency$_{\text{normalized ssODN 'x'}}$ = (Efficiency$_{\text{replicate ssODN 'x'}}$ / Efficiency$_{\text{mean control ssODN}}$) x 100%). Targeting of the *Gfp* reporter was done similarly, but on cells that were seeded one day before to 12-wells (2.8 x 10$^5$ cells/well) and by transfection that was scaled down by a factor of 2.5. 24 h after transfection cells were passaged 1:10 to 6-wells and were analyzed for *Gfp*-expression 5 days post transfection by flow cytometry on a Calibur (Becton Dickinson) or Cyan ADP (Beckman Coulter) machine. DAPI was used as a live-dead marker and cytometry data was analyzed using Summit V4.3 (Dako Colorado Inc.). ssODNs were obtained salt-free from Eurogentec and dissolved to 1 μg/μL in PBS or T$_{10}$E$_{0.1}$; a complete list with ssODN sequences, sample sizes and underlying numerical data can be found in S1 Dataset.

### Sanger sequencing of individual colonies

Analysis of the *neo* reporter or *Msh2* exon 12 sequence was done on individual G418 or 6TG resistant (G418$^R$/6TG$^R$) colonies. Colonies were picked after selection from one or more replicate experiments and expanded in 96-wells on top of irradiated MEFs. After isolation of genomic DNA, PCR was used to amplify and Sanger sequence the *neo* start region or exon 12 of

*Msh2* from each colony individually. Figures indicate the number of colonies with incorporation of corresponding AMM over the total number of sequenced colonies.

## Generation of TREX1 knock-out clones and hTREX1 overexpression

TREX1 knock-out (KO) clones were generated in WT *neo* mESCs by CRISPR/Cas9 using two guide (g)RNAs (gRNA#1 GAGCCGGAGTGCCGTACAT; gRNA#2 GACTTCGGGCCGAGA CGA) targeting the single exon of *Trex1*. gRNAs were spaced by 618 bp and were cloned into pX330.pgkpur [42,48]. Both *Cas9-Trex1*-gRNA vectors (500 ng each) were simultaneously transfected to cells growing in 6-wells by use of transfection reagent TransIT-LT1 (Mirus) [42]. Cells were passaged 24 h after transfection 1:4 to 6-wells and we selected for vector uptake with 3.6 μg/mL puromycin (Sigma-Aldrich) for 48 h. After subcloning loss of TREX1 expression was confirmed by western blot using primary a-mTREX1 (#611986, BD Biosciences), a-γ-tubulin (#T6557, Sigma-Aldrich) and secondary a-mouse IgG IRDye 800CW (#926–32210, Licor) antibodies. We generated TREX1 overexpression in WT *neo*$^{AAG}$ mESCs by lentiviral transduction of hTREX1 (pLX304-Blast-hTrex1-V5; OHS6085-101926659; Dharmacon). We selected for successful lentiviral integration by blasticidin selection at 10 μg/mL for 5 days. TREX1 overexpression in the pool of surviving cells was verified by western blot according to the protocol described above.

## Generation of FEN1 knock-down clones

FEN1 knock-down (KD) clones were independently generated in MMR⁻ *neo*$^{AAG}$ mESCs by individual lentiviral transduction of two different *Fen1* hairpins (TRC mouse library; TRCN0000071131, TRCN0000071132; Dharmacon). We selected for viral integration with 1.8 μg/mL puromycin for 3 days. After subcloning we isolated total RNA (High Pure RNA isolation kit, Roche) and used SuperScript II reverse transcriptase (Thermofisher Scientific) to prepare cDNA. *Fen1* expression was determined on cDNA by qPCR which was normalized to *β-actin* expression (qPCR primers: Fen1-Fw TTCACGGCCTTGCCAAACTAA, -Rev TGCGA CCAAAGTAGCTCTTGA; β-actin-Fw TCCACCCGCGAGCACAGCTTCTTTG, -Rev ACAT GCCGGAGCCGTTGTCGACG). FEN1 KD was validated by western blotting using primary a-FEN1 (#SC-28355, Santa Cruz Biotechnology), a-CDK4 (#SC-260-G, Santa Cruz Biotechnology) and secondary a-mouse or a-goat IgG IRDye 800CW antibodies (#926–32210, #926–32214, Licor).

## Data analysis

Statistical significance of targeting efficiencies was determined using Graphpad Prism 7 (Graphpad Software, Inc) as indicated in figure legends: two-sided t-tests, corrected for multiple testing by Holm-Sidak method if applicable; one-way or two-way ANOVA, corrected for multiple testing by Holm-Sidak method. To determine the statistical significance of AMM integration rates for different LMOs we used the two-sided Fisher's exact test using the same software. 95% confidence intervals were calculated using Wilson/Brown method to quantify the uncertainty of the calculated proportions based on compiled sequencing data. Level of significance was indicated as follows: not significant, ns; P<0.05, *; P<0.01, **; P<0.001, ***; P<0.0001, ****.

## Supporting information

**S1 Fig. Evasion of MMR-dependent suppression for AMMp5 and p9 by additional LNA modification.** (**A, B**) Sequence (A) and relative *neo* targeting efficiency in MMR⁻ and MMR⁺

cells (B) of AMMp5 LMOs with or without second LNA modification on p5. Significance was determined using a corrected multiple t-test. (**C**, **D**) Sequence (C) and relative *neo* targeting efficiency in MMR⁻ and MMR⁺ cells (D) of AMMp9 LMOs with or without second LNA modification on p9. Bars indicate the mean with SD of at least five (B) or three (D) experiments. Blue capital characters indicate mismatches with respect to the reporter, underlined characters indicate LNA modifications. Significance was determined using a corrected multiple t-test. (TIF)

**S2 Fig. Effects of a single AMM on targeting efficiency at the *Msh2* gene.** (**A**) Sequence of 5'-Acr modified antisense *Msh2* P622L LMOs with single AMMs. Blue capital characters indicate mismatches with respect to *Msh2* exon 12, underlined characters indicate LNA modifications. (**B**) Background-corrected targeting efficiencies of LMOs with a single AMM in $Msh2^{+pur/\Delta}$ cells. LMOs with an LNA on p5, 9 or 17 (+LNA) were used to mimic targeting in MMR⁻ cells. A non-specific LMO was used to determine the rate of spontaneous 6TG^R background colony formation. NA indicates not applicable. Bars indicate the mean with SD of three experiments. Significance for comparing LMOs with different AMMs was determined using a corrected one-way ANOVA; significance for comparing LMOs with and without additional LNA modification was determined using a corrected multiple t-test. (**C**, **D**) Proportion of 6TG^R colonies in which the indicated AMM was integrated after targeting with single AMM LMOs with additional LNA (+LNA, (C)) and without additional LNA (-LNA, (D)) as determined by Sanger sequencing. Error bars represent 95% confidence interval. (TIF)

**S3 Fig. Targeting efficiency and genomic integration of LMOs with multiple AMMs.** (**A**, **B**) Sequence (A) and relative *neo* targeting efficiency in MMR⁻ and MMR⁺ cells (B) of LMOs with two, three or four AMMs. Blue capital characters indicate mismatches with respect to the reporter, underlined characters indicate LNA modifications. (**C**, **D**) Frequency of AMM integration at indicated positions after targeting with LMOs from (A) in MMR⁻ (C) and MMR⁺ cells (D) as determined by Sanger sequencing. (**E**, **F**) Sequence (E) and relative *neo* targeting efficiency in MMR⁻ and MMR⁺ cells (F) of 5'-Acr modified LMOs with one, two, or three AMMs. Bars indicate the mean with SD of at least three experiments. (**G**, **H**) Frequency of AMM integration at indicated positions after targeting with 5'-Acr LMOs from (E) in MMR⁻ (G) and MMR⁺ cells (H) as determined by Sanger sequencing. Error bars (C, D, G, H) represent 95% confidence intervals. (**I, J**) LMOs with non-homologous 3'-arms (I) and targeting efficiency (L) for *Gfp* reporter in MMR⁺ cells. LMO 9xN represents a mix of LMOs in which the 3'-arm contains nine randomly introduced nucleotides. Bars indicate the mean with SD of two or three experiments. (**K, L**) 5'-Acr-modified antisense LMOs introducing the pathogenic P622L substitution in *Msh2* containing mononucleotide tracts in the 3'-arm (K) and corrected targeting efficiencies (L). Bars indicate the mean with SD of four experiments. Significance was determined using a corrected two-way (B, F) or one-way ANOVA (J, L). (TIF)

**S4 Fig. Sanger sequencing data from colonies modified by LMOs with multiple AMMs.** (**A**, **B**) Sequencing data from individual colonies (rows) modified by LMOs with multiple AMMs in MMR⁻ (A) and MMR⁺ (B) cells. (**C**, **D**) Data from cells targeted with 5'-Acr modified LMOs with one or multiple AMMs in MMR⁻ (C) and MMR⁺ (D) cells. Total integration frequencies per position are presented in S3 Fig. (TIF)

**S5 Fig. Targeting efficiency and genomic integration of LMOs with AMMp5 in *Fen1* KD clones.** (**A, B**) Quantification of *Fen1* gene expression by RT-qPCR (A) and FEN1 protein

levels by western blot (B) in two independent stable *Fen1* KD clones generated by lentiviral shRNA integration in MMR⁻ mESCs with *neo* reporter. *Fen1* expression was normalized against *β-actin* expression and data was obtained in two experiments with two technical replicates; bars indicate mean with SD. Significance was determined using a corrected one-way ANOVA. (**C**) Relative *neo* targeting efficiency using a 5'-Acr modified control and AMMp5 LMO in parental and *Fen1* KD clones A1 and B4. Efficiency was normalized against efficiency obtained with the control LMO in the parental cell line. Bars indicate the mean with SD of at least two experiments. Significance was determined using a corrected two-way ANOVA. (**D**) Proportion of G418$^R$ colonies with integration of AMMp5 in MMR⁻ *Fen1* KD clones A1 and B4 after targeting with 5'-Acr modified AMMp5 LMO as determined by Sanger sequencing. (**E**) Relative neo targeting efficiency of 5'-phosphate modified LMOs in the presence and absence of AMMp5; data from at least two experiments. Significance was determined using a corrected one-way ANOVA. (**F**) Proportion of MMR⁻ G418$^R$ cells in which AMMp5 was integrated. Error bars in (D) and (F) represent 95% confidence interval.
(TIF)

**S6 Fig. Protection of ssODNs from 3'-5' degradation by various modifications in the 3'-arm.** (**A**) Sequences of ssODNs (without LNA modification of the AAG-correcting central nucleotide) with AMMp17 and various 3'-arm modifications. Blue capital characters indicate mismatches with respect to the *neo* reporter, underlined characters indicate LNA modifications, '1' indicates internal modification with 6-chloro-2-methoxyacridine, red underlined capital characters indicate modification with 2'O-Methyl nucleotides and asterisks indicate PTO-modified bonds. (**B**) Relative *neo* targeting efficiency in MMR⁻ cells with AMMp17 containing ssODNs in combination with 3'-arm modifications. Bars indicate the mean and SD from at least three experiments. Significance was determined using a corrected one-way ANOVA. (**C**) Proportion of MMR⁻ G418$^R$ colonies in which AMMp17 was integrated. Error bars represent 95% confidence interval.
(TIF)

**S7 Fig. Targeting efficiency in TREX1 KO and hTREX1 overexpressing mESCs.** (**A**) Confirmation of CRISPR/Cas9 mediated knockout of TREX1 in MMR⁺ mESCs by western blot. (**B**) Efficiency of *neo* targeting in four TREX1 KO clones with 400 pmol 25 nt LMOs. Bars indicate the mean with SD of at least three experiments. Significance was determined using a corrected one-way ANOVA. (**C, D**) Validation (C) and quantification (D) of hTREX1 overexpression (OE) in MMR⁺ mESCs by western blot. Bars indicate mean and SD from two experiments. (**E**) *Neo* targeting efficiency with 5'-Acr modified LMO in MMR⁺ parental and TREX1 OE cells. Bars indicate mean and SD from four experiments. Significance was determined using a student's t-test.
(TIF)

**S8 Fig. LNA modification improves targeting efficiency in MMR⁻ cells if mismatch is placed centrally or in 3'-arm.** Relative *Gfp* targeting efficiency of ssODNs in presence and absence of LNA modification on the AAG-correcting mismatch in MMR⁻ cells. Efficiency was normalized to ssODN with centrally positioned mismatch (p13) without LNA. Bars indicate mean with SD from seven experiments. Significance was determined using a corrected two-way ANOVA.
(TIF)

**S1 Dataset. LMO sequences and numerical data.**
(XLSX)

## Acknowledgments

We are thankful for the technical assistance from Melis Bartsch with generating the TREX1 knockout cell lines.

## Author Contributions

**Conceptualization:** Thomas W. van Ravesteyn, Hein te Riele.

**Data curation:** Thomas W. van Ravesteyn.

**Formal analysis:** Thomas W. van Ravesteyn.

**Funding acquisition:** Hein te Riele.

**Investigation:** Thomas W. van Ravesteyn, Marcos Arranz Dols, Wietske Pieters, Marleen Dekker, Hein te Riele.

**Methodology:** Thomas W. van Ravesteyn.

**Supervision:** Hein te Riele.

**Writing – original draft:** Thomas W. van Ravesteyn, Hein te Riele.

**Writing – review & editing:** Thomas W. van Ravesteyn, Hein te Riele.

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
