## [Decision Letter · Decision Letter 0]

22 Feb 2020

Dear Dr te Riele,

Thank you very much for submitting your Research Article entitled 'Extensive trimming of short single-stranded DNA oligonucleotides during replication-coupled gene editing in mammalian cells' to PLOS Genetics. Your manuscript was fully evaluated at the editorial level and by independent peer reviewers. The reviewers appreciated the attention to an important problem, but raised some substantial concerns about the current manuscript. Based on the reviews, we will not be able to accept this version of the manuscript, but we would be willing to review again a much-revised version. We cannot, of course, promise publication at that time.

Should you decide to revise the manuscript for further consideration here, your revisions should address the specific points made by each reviewer. We will also require a detailed list of your responses to the review comments and a description of the changes you have made in the manuscript.  It  would be mostly helpful to the next round of reviews, if you upload a marked up copy showing all changes as a separate file.

If you decide to revise the manuscript for further consideration at PLOS Genetics, please aim to resubmit within the next 60 days, unless it will take extra time to address the concerns of the reviewers, in which case we would appreciate an expected resubmission date by email to plosgenetics@plos.org.

[LINK]

We are sorry that we cannot be more positive about your manuscript at this stage. Please do not hesitate to contact us if you have any concerns or questions.

Yours sincerely,

Dmitry A. Gordenin

Associate Editor

PLOS Genetics

Gregory Barsh

Editor-in-Chief

PLOS Genetics

Reviewer's Responses to Questions

**Comments to the Authors:**

Reviewer #1: The study uses locked nucleic acid (LNA)-modified single-strand oligonucleotides (LMOs) in gene editing in mammalian cells, mouse embryonic stem cells (mESCs), without induction of a break in the target DNA. The work shows that the 3’-end of the LMOs is trimmed before being integrated into cell DNA during DNA replication, while the 5’-end is trimmed after integration. Mutations in the 3’-arm of the LMO are resistant to cellular mismatch repair (MMR).

Recent work by the authors have found that an LNA modification in the central mismatch nucleotide of an ssODN is not targeted by cellular MMR in mESCs and E. coli. LMOs 25-nt long are used to correct a defective start codon of a neomycin reporter gene integrated in the Rosa26 locus of MMR+ and MMR- mESCs. Gene correction is measured by frequency of G418-resistant colonies. It is found that the 3’-arm sequence does not need to be complementary to the target (used stretch of As, Cs, or Ts; stretch of Gs was inhibitory, possibly due to formation of secondary structures). It is not clear how the 5’ end is processed. While the 3’ could be processed by TREX1 exonuclease 3’to 5’-exonuclease.

Truncations of the 5’ end reduced the efficiency of gene editing by the LMOs; while this was less evident for truncations at the 3’-arm, although the highest efficiency was observed in the 25-mer LMO when the mutation was in the center of the molecule.

The study is well written, experiments are planned well, and the results are of significant interest in the gene editing/DNA repair community. However, the value of the LMO technology is not immediately clear because frequencies of gene editing are not evident, and the risk of off-target DNA modifications is not taken into consideration.

Major points:

- In the Results, it will be valuable to provide what is the actual frequency of gene editing in the mESC using the LMOs for the correction of the neomycin reporter gene. Is this frequency sufficiently high for editing DNA in cases in which there is no change in phenotype after the gene correction? How does LMO technology with no break in the target locus compare to gene editing following DNA break induction? Considering the results of this study, it will be important to clearly discuss what can be most significant applications of the LMO technology.

- Showing Western blot data for FEN1 KD would be a much stronger result than RT-PCR. RT-PCR does not provide a measure of the protein level in the cells. Thus, in this study, it cannot be concluded that FEN1 has no effect on gene editing by LMOs in mESCs.

- If most of the 3’-end of the LMOs can be of any sequence (NNNNNN) and only ~ 8nt of the 25mer LMOs are integrated in the DNA at the site of gene editing, the LMOs can likely recognize multiple targets, in part like miRNAs do. Did the authors examined whether the LMOs are integrated at other sites? Does this small size of complementarity with the target sequence represent a risk for off-target editing? It would be valuable if the authors could also discuss the value of LMO-driven gene editing with respect to its risk of off-target modifications.

Other comments:

-The authors write on page 5: <<by -proficient="" and="" cells="" comparing="" efficiencies="" in="" mmr-deficient="" normalized="" the="">>. It would be helpful to explain what does “normalized efficiencies” mean in this context, and exactly what was done to normalize the data.

-In Figure 1C and D (and other Suppl figures), it is not clear what is the meaning of the confidence interval. Were the samples sequenced multiple times, how many times? How many colonies were sequenced from each individual transfection experiment? For example, 24/24 does it meant that there were 24 G418 colonies among the 3 repeats (shown in Fig. 1B) and all of them were sequenced, how many from each of the 3 transfections?

-Did the authors examine whether random sequence, which is not a homonucleotide run, in the 3’-arm also has no impact of gene editing like multiple As, Cs, and Ts?

-Should correct in Figure S7D Fen1 to Trex1 in the legend of the Y axis.

Reviewer #2: attached.

Reviewer #3: The manuscript by van Ravesteyn et al describes studies to examine the fate of DNA base mismatches formed during replication-coupled oligonucleotide gene editing in mouse embryonic stem cells (mESCs). This lab had previously shown that a locked nucleic acid (LNA) was able to confer resistance to the mismatch repair (MMR) system of mESCs, allowing them to achieve higher frequencies of oligo-mediated mutagenesis in MMR+ backgrounds. Their assay consists of a centrally placed nucleotide in the oligos, which when successfully transferred to the reporter construct following transformation, establishes a start codon for expression of a chromosomally-encoded neomycin resistant gene. The nucleotide restoring a start codon to the neo gene is an LNA, making it resistant to MMR.

In this report, they use the same assay to examine if alternate mismatches (AMM , +/- LNAs) in the 5’ and 3’ arms of the oligos survive long enough to become incorporated into the chromosome. By following the effects of the AMMs on the efficiency of NeoR colony formation, and what frequency the bases forming the AMM get transferred to the recombinant, the authors have made conclusions regarding the processing of the oligo both before and after it anneals to the replication fork.

The authors show that AMMs in the 5’ arm of their oligos suppress targeting efficiencies of the neo-reporter construct, and that this suppression is exacerbated by the action of the MMR system. That there is an effect of MMR suggests that the 5’ arm anneals to the target. It is noted that the bases in the 5’ arm of the oligo that create the AMM (except AMMp9) are not present in the NeoR colonies following sequencing. This result is true even in MMR-deficient host, suggesting that these bases are not lost due to MMR, but to nucleolytic processing. Furthermore, they show that when the 5’ arm is protected from degradation, increased recombinant formation is observed, though there is still lack of uptake of the first six-to-eight 5’ AMMs in the recombinant. The result suggests endonucleolytic processing in the 5’ arm of the oligo, while annealed to its target. This was further evidenced by the dramatic increase in AMMp5 incorporation by including three phophothionate linkages on either side of AMMp5 in the oligo.

This situation contrasts with the 3’ arm of the oligo where MMR has no effect on targeting efficiency and where the sequence composition is largely irrelevant to recombinant formation. Based on the ability of an exonuclease resistant LNA at p21 to decrease uptake of a mismatch at p17 suggest that exonucleolytic degradation of the 3’ arm may actually favor recombinant formation. This supposition is supported by the decreased recombinant formation when a prominent host 3’-5’ ssDNA exonuclease is knocked out.

The authors have taken advantage of their Neo-reporter system for oligo-mutagenesis in mESCs to examine processing of SNP-targeting oligo in vivo. These questions have been addressed before in both E. coli and yeast, and have yielded insights into the mechanisms of oligo-mediated mutagenesis in those systems. As far as I know, this is the first examination of the processing of oligos used for gene targeting in mammalian cells, and so, is informative. In yeast, the integration patterns of AMM was unaffected by MMR (similar to what was found here), suggesting a unified role of nucleases in limiting transfer of oligos to just a small central core of bases.

The figures are well drawn, and as in previous publications from this lab, having the sequences of the oligos in each figure, with designations of which ones contain LNAs and/or create mismatches, is very helpful. Designations in the figures of whether sequencing results are from MMR+ or MMR- hosts, while in the legends, would also help.

Minor points

1. Why is the AMMp9 position one that is taken up by the colonies to such a high degree? On might argue that access is limited by the processivity of an exonuclease to that position, but the model favored by the authors is one of endonucleolytic processing of the 5’ arm. Might its proximity to the central LNA play a role in protection from degradation?

2. Line 106 “we found that MMR aggravated the suppressive effect of 5’-arm AMMs at p5”….

This sentence is a little unclear. May I suggest “…we found that the presence of MMR exacerbated the suppressive effect of 5’-arm AMMs at p5….”

Also, perhaps the authors could discuss a mechanism for this observation. One might think that MMR activity on the AMM results in collateral damage of removing the LNA-containing targeting base, but wouldn’t that base be resistant to nucleases involved in MMR?

3. Line 256 “Nevertheless, also in E. coli ssODNs showed a pattern of incomplete incorporation “

Chang to: Nevertheless, E. coli ssODNs also showed a pattern of incomplete incorporation

4. Line 166-167 “Also the targeting-suppressive effect of AMMp5 in MMR- and even more in MMR+ cells was not affected by LNA at p1 or 5’-Acr modification of the LMO.”

It would help the reader to tell them to compare the y axes of Fig 3B and C.</by>

**Have all data underlying the figures and results presented in the manuscript been provided?**

Reviewer #1: Yes

Reviewer #2: Yes

Reviewer #3: Yes

PLOS authors have the option to publish the peer review history of their article (what does this mean?). If published, this will include your full peer review and any attached files.

Reviewer #1: No

Reviewer #2: No

Reviewer #3: No

---

## [Decision Letter · Decision Letter 1]

10 Aug 2020

Dear Dr te Riele,

We are pleased to inform you that your manuscript entitled "Extensive trimming of short single-stranded DNA oligonucleotides during replication-coupled gene editing in mammalian cells" has been editorially accepted for publication in PLOS Genetics. Congratulations!

Before your submission can be formally accepted and sent to production you will need to complete our formatting changes, which you will receive in a follow up email. Also, even if not listed in that email, please define VUS abbreviation as requested by Reviewer 3.  Please be aware that it may take several days for you to receive this email; during this time no action is required by you. Please note: the accept date on your published article will reflect the date of this provisional accept, but your manuscript will not be scheduled for publication until the required changes have been made.

Yours sincerely,

Dmitry A. Gordenin

Associate Editor

PLOS Genetics

Gregory Barsh

Editor-in-Chief

PLOS Genetics

Comments from the reviewers (if applicable):

Reviewer's Responses to Questions

**Comments to the Authors:**

Reviewer #1: The Authors have address the concerns of this reviewer and added additional experiments that strengthened the study.

Reviewer #3: The authors have addressed my concerns from the previous review.

One thought came to mind in rereading the manuscript. Might there not be role for the proofreading function of the replicative polymerase in removing the AMMs in the 3' arm? While the experiment of using the non- annealing homopolymers for the 3' arm argues FOR digestion by an non-replisome 3'-5' exonuclease, it does not rule out the proofreading capability of DNA polymerase acting in cases where annealing can take place.

Along these lines, is it know if an LNA in the context of a lagging strand is resistant to proofreading functions?

If so, that might explain the protective effects of the LNA for AMMP17 incorporation (lines 208-210).

On a minor point, the term VUS is explained in the authors response, but it is not defined in the text of the manuscript. It first appears in the manuscript text in line 324, but it had not yet been defined.

**Have all data underlying the figures and results presented in the manuscript been provided?**

Reviewer #1: None

Reviewer #3: Yes

PLOS authors have the option to publish the peer review history of their article (what does this mean?). If published, this will include your full peer review and any attached files.

Reviewer #1: No

Reviewer #3: No

**Data Deposition**

http://datadryad.org/submit?journalID=pgenetics&manu=PGENETICS-D-20-00005R1

**Press Queries**

---

## [Editor Report · Acceptance letter]

7 Oct 2020

PGENETICS-D-20-00005R1 

Extensive trimming of short single-stranded DNA oligonucleotides during replication-coupled gene editing in mammalian cells 

Dear Dr te Riele, 

We are pleased to inform you that your manuscript entitled "Extensive trimming of short single-stranded DNA oligonucleotides during replication-coupled gene editing in mammalian cells" has been formally accepted for publication in PLOS Genetics! Your manuscript is now with our production department and you will be notified of the publication date in due course.

With kind regards,

Matt Lyles

PLOS Genetics

On behalf of:
